

# Extraction, identification and component analysis of exosome-like nanovesicles in *Anoectochilus roxburghii* (Wall.) Lindl.

Xuanzhe Zhou[1], Xinglin Ruan[2] and Xue Mi[3]

[1] School of Basic Medical Sciences, Fujian Medical University, Fuzhou, Fujian, China
[2] Department of Neurology, Fujian Medical University Union Hospital, Fuzhou, Fujian, China
[3] Public Technology Service Center, Fujian Medical University, Fuzhou, Fujian, China

## ABSTRACT

**Objective**. This study aimed to investigate the presence of exosome-like nanovesicles in *Anoectochilus roxburghii* (Wall.) Lindl. (*A. roxburghii*) and the related active ingredients.

**Methods**. *A. roxburghii*-derived exosome-like nanovesicles (ARENVs) were extracted by the differential ultracentrifugation and characterized by transmission electron microscopy and nano-flow cytometry. The components in ARENVs derived from soil-cultured and hydroponic *A. roxburghii* were identified and quantified by liquid chromatography-mass spectrometry.

**Results**. Nanovesicles with typical exosome characteristics were successfully isolated from *A. roxburghii*. After analysis and identification, ARENVs were rich in multiple flavonoid components such as isoquercitrin, kaempferol, isorhamnetin, narcissoside, quercetin and rutin. The quantity of flavonoid compounds was higher in ARENVs derived from soil-cultured *A. roxburghii* than in those from hydroponic *A. roxburghii*.

**Conclusion**. *A. roxburghii* contains exosome-like nanovesicles, which are rich in active flavonoid ingredients. These active ingredients are more abundant in ARENVs derived from soil-cultured *A. roxburghii* than hydroponic counterparts. The findings may provide an innovative theoretical basis for further exploration of the medicinal value of *A. roxburghii*.

Corresponding authors
Xinglin Ruan, xlruan@163.com
Xue Mi, mixue0508@163.com

## INTRODUCTION

*Anoectochilus roxburghii* (Wall.) Lindl. (*A. roxburghii*) is a perennial herbaceous plant of the Orchidaceous family, with its leaves featuring a network of golden veins and a unique shape. The plant is widely distributed in subtropical to tropical regions such as southeastern China, India, Nepal and Southeast Asia, and known as the ''king of medicine'' (*Liu et al., 2015*). Current phytochemical studies have shown that the active ingredients of *A. roxburghii* mainly include flavonoids (such as quercetin, isoquercitrin, isorhamnetin), polysaccharides, alkaloids and terpenoids (*Zou et al., 2024*). These constituents exhibit anti-inflammatory (*Wang et al., 2025*), antioxidant (*Xiao et al., 2024*), anti-tumor (*Yu et al., 2017*; *Gunes et al., 2023*), and hypoglycemic (*Tian et al., 2022*) properties, which can

provide liver protection (*Dong et al., 2025*; *Yu et al., 2021*; *Lin et al., 2024*; *Huang et al., 2023*), kidney protection (*Li et al., 2016*) and relief of cognitive impairment (*Xiao et al., 2024*; *Fu et al., 2022a*). Therefore, it is worthwhile to explore its therapeutic values in clinical settings.

Wild *A. roxburghii* has extremely demanding growing environments and has become an endangered Chinese medicinal herb. Currently, most *A. roxburghii* available on the market is cultivated. Among these cultivation methods, soil-based cultivation is the most traditional and authentically natural. Under these conditions, *A. roxburghii* thrives in a complex soil matrix composed of minerals, organic matter, water, air, and microorganisms. On the other hand, hydroponics, another important cultivation method, provides a controlled environment with substantial nutrition supply but low microbial contamination for plant growth (*Fussy & Papenbrock, 2022*). Notably, the medicinal value of *A. roxburghii* depends primarily on its secondary metabolites (such as flavonoids and polysaccharides), whose synthesis and accumulation are profoundly regulated by environmental factors (*Yang et al., 2018*), including nutrient availability, water conditions, light, temperature, and particularly biotic and abiotic stresses (*Ye et al., 2017*; *Ye et al., 2020*; *Lv et al., 2024*; *Ramakrishna & Ravishankar, 2011*). Therefore, soil cultivation and hydroponics offer distinct root environments and nutrient acquisition methods, which are likely to significantly influence the physiological state of *A. roxburghii*, particularly its secondary metabolic pathways, product types, and quantity.

Molecularly, cells can secrete exosomes that are nanoscale membrane vesicles with a diameter of about 30–150 nm. The components of exosomes may contain beneficial agents from the parental cells. In recent years, plant-derived exosome-like nanoparticles (PENs) have attracted extensive attention as a new system for the delivery of active ingredients (*Zhao et al., 2024*; *Kim et al., 2022*). Studies have shown that PENs from plants such as *Panax ginseng* C.A. Meyer (ginseng) (*Kim et al., 2023*), *Lonicera japonica* Thunb. (*Chi et al., 2024*), *Pueraria lobata* (Willd.) Ohwi (*Lu et al., 2024*), *Zingiber officinale* Roscoe (ginger) (*Xie et al., 2024*), and *Morus nigra* L. leaves (*Gao et al., 2024*) are rich in lipids, miRNAs, and functional proteins. These ingredients can mediate intercellular communication and engage in biological activities, delivering antioxidant, antiviral, anti-inflammatory effects and participating in intestinal barrier repair and immunomodulation. However, it remains unexplored whether such exosome-like nanovesicles exist in *A. roxburghii* and what components they contain.

This study aimed to determine whether exosome-like nanovesicles (ARENVs) exist in *A. roxburghii*, explore whether these vesicles contain bioactive components, and compare the differences in bioactive components in ARENVs derived from soil-cultured and hydroponic *A. roxburghii*. The findings may shed novel lights on the research and application of *A. roxburghii*, and further promote the in-depth development and rational utilization of *A. roxburghii* in clinical treatments.

## MATERIALS AND METHODS

### Materials, reagents and standards

Fresh samples of soil-cultured and hydroponic *A. roxburghii* were obtained from a planting base (Fujian Huazhiyun Biotechnology Co., Ltd, Yong'an, Fujian Province, China). A vouchered specimen (25415) was preserved in our laboratory. Uranyl acetate was purchased from Zhongjingkeyi Film Technology Co., Ltd (Beijing, China). D-mannitol was acquired from Sangon Biotech Co., Ltd (Beijing, China). Pectinase, cellulase, and the standard substances used in the experiment, including rutin, quercetin, isorhamnetin, isoquercitrin, kaempferol, and narcissoside, were obtained from Shanghai Yuanye BioTechnology Co., Ltd. (Shanghai, China).

### Isolation of ARENVs

The exosome-like nanoparticles were extracted from *A. roxburghii* and characterized, as previously reported with slight modification (*Zhao et al., 2023*). Healthy specimens of soil-cultured and hydroponic *A. roxburghii* were selected, then meticulously cleaned with deionized water, and minced. Subsequently, the minced *A. roxburghii* (250 grams) was immersed into mixed enzyme solution (4% cellulase, 2% pectinase, 0.6 mol/L mannitol; pH 5.8) and underwent enzymolysis at 50 °C for 6 h. The enzymolysis solution was then centrifuged at 4 °C and $16,000\times$ g for 60 min. Next, the supernatant was successively centrifuged at 4 °C and $2,000\times$ g for 30 min and at 4 °C and $10,000\times$ g for 45 min. Subsequently, the supernatant was filtered through a 0.45 $\mu$m filter membrane and the filtrate was transferred to an ultracentrifuge tube and centrifuged at 4 °C and $100,000\times$ g for 70 min. The supernatant was removed and the precipitate were resuspended in 10 mL of pre-cooled phosphate buffer solution (PBS) and ultracentrifuged again at 4 °C and $100,000\times$ g for 70 min. Finally, the precipitate, designated as ARENVs, was resuspended in 550 $\mu$L of pre-cooled PBS and stored at $-80$ °C until use.

### Analysis of ARENVs by transmission electron microscopy

Transmission electron microscopy (TEM) analysis was performed as previously described in *Zhao et al. (2023)* with slight modification. A volume of 10 $\mu$L of ARENVs sample was gently placed on a formvar-coated copper grid. Next, 2% uranyl acetate was applied and remained undisturbed for 5 min. All steps were carried out in the dark. After that, the sample was then dried for imaging, and the image was obtained using a Tecnai G2 TEM.

### Particle size distribution and concentration analysis

A nano-flow cytometer was used to analyzed the particle size and concentration of ARENVs. According to the instructions, the instrument performance was first tested with a standard sample. After passing the test, a volume of 10 $\mu$L of ARENVs was diluted to an appropriate concentration and then tested on the instrument. The particle size distribution and concentration were calculated with the NanoFCM software NF professional v1.08.

### Protein extraction and concentration determination

A volume of 20 $\mu$L of ARENVs was obtained and quickly mixed with a $5\times$ RIPA lysis buffer. The mixture was lysed on ice for 30 min. Afterwards, a volume of 20 $\mu$L of the

mixture was retrieved for the determination of the protein content in ARENVs with the BCA protein quantification kit.

## LC-MS/MS analysis

According to the method outlined in references (*Mi et al., 2025*; *Zhang et al., 2022*), The concentration of six flavonoids compounds of *A. roxburghii* within the ARENVs was determined by triple quadrupole liquid chromatography-mass spectrometry (LC-QqQ MS/MS).

Regarding preparation of standard solutions, the reference standards of rutin, narcissoside, quercetin, kaempferol, isoquercitrin, isorhamnetin were precisely weighed, prepared into a standard stock solution at a concentration of 1 mg/mL with methanol, and stored at −20 °C. A mixed stock solution was created by combining the six analytes in suitable ratios, then diluted with methanol to produce a range of concentrations for the working solution. It was stored at 4 °C, filtered through a 0.22 μm needle filter before the injection into the liquid chromatography-tandem mass spectrometry (LC-MS/MS) system.

For sample preparation, a volume of 200 μL of methanol was added to 50 μL of ARENVs sample, vortexed for 60 s, and then centrifuged at 12,000 rpm and 4 °C for 10 min. The supernatant was transferred to a new EP tube and centrifuged again for 10 min under the above conditions. The resulting supernatant was transferred to a liquid-phase chromatography vial for subsequent analysis.

Chromatographic and mass spectrometry conditions were set following a previous method with modifications (*Zhang et al., 2022*). The examination of the samples was conducted on a Shimadzu UFLC-MS/MS 8040 system, which was equipped with an LC-20AD pump, a SIL-20AC autosampler, an FCV-20A controller, and a CTO-20A column oven. An Agilent-C18 (2.1 mm ×100 mm, 3.5 μm) was used for the chromatographic separation. The mobile phase consisted of water with 0.1% (v/v) formic acid and 10 mM ammonium acetate (A) and methyl alcohol (B). The flow rate was kept at 0.3 mL/min, with the column oven temperature held at 35 °C and an injection volume of five μL. The chromatographic gradient used in this experiment was: 0–0.5 min, 90% A; 0.5–4 min, a linear change from 90% A to 10% A; 4–4.5 min, 10% A; 4.5–12 min, 90% A. During the mass spectrometry detection, the electrospray ionization (ESI) interface was used to detect the analytes in the negative ion mode. The analytes were then quantified by the multiple reaction monitoring (MRM) mode. For the ESI settings, the heating block was set at 400 °C, the desolvation line (DL) was keptat 250 °C, and the drying gas ($N_2$) and the flow rates of the nebulizing gas ($N_2$) were at 14 L/min and 3 L/min, respectively.

Sample determination was performed as previously described in *Mi et al. (2025)*. Specifically, the MRM transition and retention time were used to quantify and identify the compounds. Based on the detected signal, the one with a higher signal value and clearer mass spectra was chosen for quantitative analysis. The external standard calibration curves were used to calculate the concentrations of the six selected analytes.

## Statistical analysis

Data were presented as mean ± standard error of the mean (SEM) and statistically analyzed with GraphPad Prism 9.0 software (GraphPad Prism Software Inc., USA). The differences

between groups were assessed by the Unpaired two-sample $t$-test. The statistical difference was designated at $p < 0.05$.

## RESULTS

### Morphological identification of ARENVs

ARENVs were extracted by ultracentrifugation. The schematic diagram is shown in Fig. 1. TEM observation showed that the vesicles extracted from *A. roxburghii* featured a typical cup-shaped or saucer-shaped structure (Fig. 2A), which was consistent with the morphological characteristics of exosomes, preliminarily proving the existence of exosome-like nanovesicles in *A. roxburghii.*

### Particle size analysis

The results of nano-flow cytometry (nFCM) showed that the particle size of ARENVs mainly varied between 50 and 150 nm (Fig. 2B), with an average particle size of 85.7 nm and a concentration of 2.32E + 10 /mL, which further verifies that the extracted nanovesicles are exosomes.

### Determination of protein concentration

The quantification of BCA protein showed that the protein concentration of ARENVs was $5.9 \pm 0.3$ mg/mL. These proteins may include membrane structure-related proteins of ARENVs, proteins that maintain the skeleton and vesicle formation, and biologically active protein molecules in the vesicles.

### Quantitative analysis of flavonoid compounds in ARENVs

Flavonoid compounds are the representative bioactive ingredient of *A. roxburghii.* To ascertain whether ARENVs contain any flavonoid compounds, six classical active components, including rutin, narcissoside, quercetin, kaempferol, isoquercitrin, isorhamnetin, were quantified by LC-MS/MS. Table 1 displays the MRM transition and retention time for the six compounds, and Fig. 3 illustrates the representative chromatograms of the extracted ions in standard solutions. The standard curve along with the linear regression equation, and correlation coefficient is presented in Fig. S1. The content of compounds in three different batches of ARENVs was determined, with the results illustrated in Fig. 4. Our results showed that ARENVs were rich in flavonoid active ingredients, especially isoquercitrin and kaempferol, which were higher in soil-based *A. roxburghii.* In the ARENVs obtained by soil-based cultivation and hydroponics, the content of rutin was $(29.52 \pm 6.61)$ ng/100 g and $(3.54 \pm 0.58)$ ng/100 g, respectively; the content of isoquercitrin was $(1{,}096.81 \pm 126.12)$ ng/100 g and $(14.71 \pm 1.38)$ ng/100 g, respectively; that of narcissoside was $(116.33 \pm 3.75)$ ng/100 g and $(33.68 \pm 3.21)$ ng/100 g, respectively; that of quercetin was $(106.94 \pm 2.46)$ ng/100 g and $(4.06 \pm 0.19)$ ng/100 g, respectively; that of kaempferol was $(353.89 \pm 49.39)$ ng/100 g and $(2.85 \pm 1.74)$ ng/100 g; and that of isorhamnetin was $(143.082 \pm 20.77)$ ng/100 g and $(14.71 \pm 1.38)$ ng/100 g, respectively. The contents of these six compounds were higher in ARENVs of soil-cultured *A. roxburghii* than in those of hydroponic culture, which, to some extent, indicates that soil-cultured *A. roxburghii* is more nutritionally valuable than hydroponically-cultured counterpart.

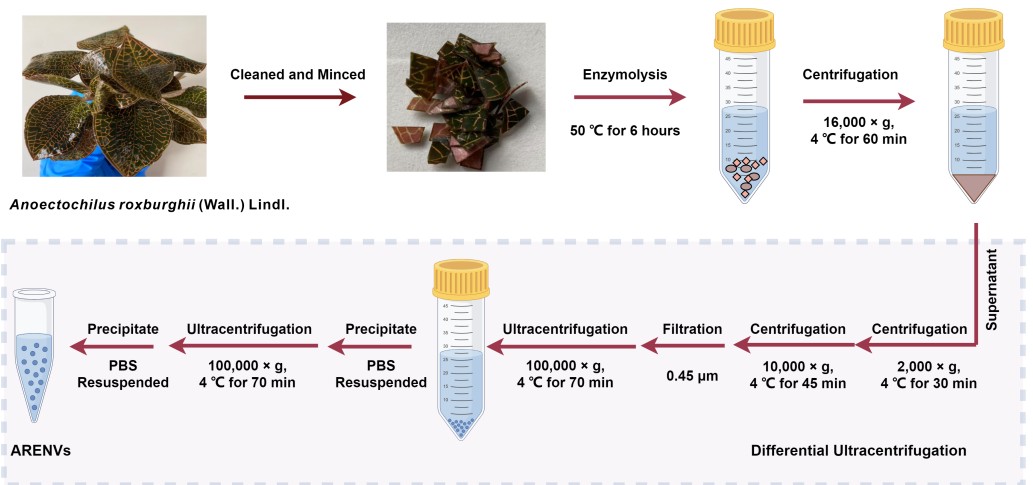

**Figure 1** **The isolation method of ARENVs by differential centrifugation (by https://www.figdraw.com/).**

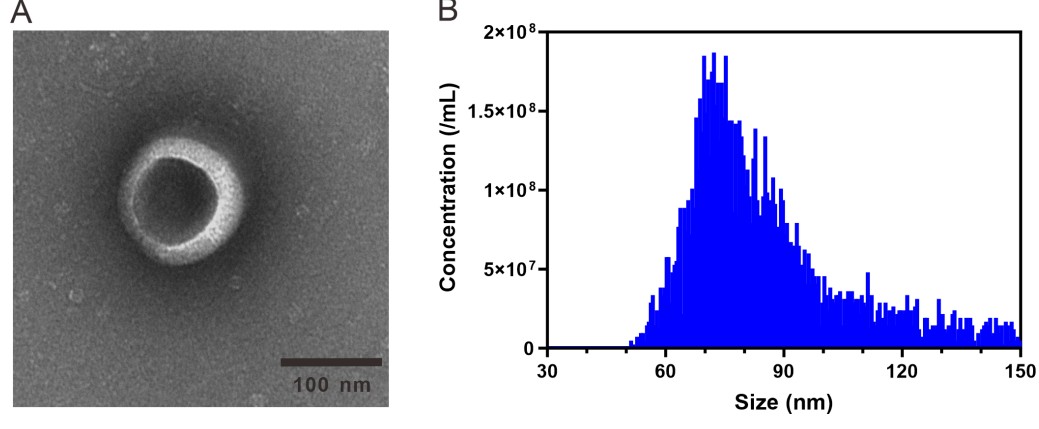

**Figure 2** **TEM images and nanoparticle size analysis of ARENVs.** (A) Morphology of ARENVs by transmission electron microscopy (TEM). The scale bar represents 100 nm. (B) The nanoparticle size analysis of ARENVs by nano-flow cytometry (nFCM).

## DISCUSSION

This study, for the first time, successfully extracted and identified nanovesicles with typical exosome characteristics from *A. roxburghii.* The cup-shaped morphology observed by TEM and the specific particle size distribution detected by nFCM confirmed the presence of exosome-like nanovesicles in *A. roxburghii.* The LC-MS/MS analysis confirmed that ARENVs contained flavonoid secondary metabolites, including isoquercitrin, kaempferol, isorhamnetin, narcissoside, quercetin and rutin. These active ingredients are more abundant in ARENVs derived from soil-cultured *A. roxburghii* than in those from the hydroponic one. This discovery provides new insights into ARENVs research and suggests

**Table 1  Identification and quantitative information of flavonoid compounds of *A. roxburghii* in ARENVs.**

| Components | CAS | Formula | MW | Mode | RT (min) | Precursor ion | product ion | CE |
|---|---|---|---|---|---|---|---|---|
| Rutin | 153-18-4 | $C_{27}H_{30}O_{16}$ | 610.52 | $[M-H]^-$ | 5.504 | 609.15 | 300.20 | 28 |
| Isoquercitrin | 482-35-9 | $C_{21}H_{20}O_{12}$ | 464.1 | $[M-H]^-$ | 5.540 | 463.10 | 300.10 | 30 |
| Narcissoside | 640-80-8 | $C_{28}H_{32}O_{16}$ | 624.551 | $[M-H]^-$ | 5.738 | 623.15 | 315.20 | 14 |
| Quercetin | 117-39-5 | $C_{15}H_{10}O_7$ | 302.24 | $[M-H]^-$ | 6.041 | 301.15 | 151.20 | 14 |
| Kaempferol | 520-18-3 | $C_{15}H_{10}O_6$ | 286.24 | $[M-H]^-$ | 6.304 | 285.05 | 93.20 | 17 |
| Isorhamnetin | 480-19-3 | $C_{16}H_{12}O_7$ | 316.267 | $[M-H]^-$ | 6.350 | 315.15 | 151.05 | 14 |

that the cultivation environment may be a key factor influencing the composition of their active ingredients.

Extracellular vesicles (EVs) are membrane-contained vesicles released by cells. They are highly conserved across prokaryotes and eukaryotes, and can transmit information to other cells and influence their function (*Yanez-Mo et al., 2015*). Exosomes are one of the main types of EVs. Vesicles of 50–200 nm have been extracted from *Helianthus annuus* L. (sunflower) seeds and been found to contain a small GTPase Rab (*Regente et al., 2009*), which engages in vesicle transport. The finding demonstrates the presence of exosome-like nanovesicles in sunflower and vesicle-based information transfer in plants. Meanwhile, PENs participate not only in the normal vesicle transport but also in the responses to biotic stresses (*Samuel et al., 2015*). For example, a large number of vesicle-like structures accumulate at the cell wall appositions of *Hordeum vulgare* L. (barley) leaves, in which they prevent the penetration attempt of the biotrophic powdery mildew fungus *Blumeria graminis* f. sp. *hordei* (*An et al., 2006b*) and the hypersensitive cell death by blocking plasmodesmata (*An et al., 2006a*). In our study, we also found exosome-like nanovesicles in *A. roxburghii*, which are rich in flavonoids. The latter plays a key role in plants' response to environmental stresses such as UV damage and pathogen invasion (*Treutter, 2005*). These findings suggest that these vesicles may also be an important component of the defense system of *A. roxburghii*, which provides a new research target for understanding the ecological adapttation of *A. roxburghii*, although the specific biological mechanism needs further investigation.

Currently, many plants have been documented to contain exosome-like nanovesicles, mainly TCM herbs (such as *Ganoderma lucidum* (Curtis) P. Karst. (*G. lucidum*) (*Mi et al., 2025*), ginseng (*Kim et al., 2023*), *Lonicera japonica* Thunb. (*Chi et al., 2024*), *Pueraria lobata* (Willd.) Ohwi (*Lu et al., 2024*), *Morus nigra* L. leaves (*Gao et al., 2024*)), and edible plants (such as *Malus domestica* Borkh (apple) (*Trentini et al., 2024*), ginger (*Xie et al., 2024*), *Citrus × paradisi* Macfad. (grapefruit) and *Solanum lycopersicum* L. (tomato) (*Kilasoniya et al., 2023*)). PENs have been reported to be rich in functional molecules such as proteins, lipids, and mRNA, potentially playing a protective role in plants. For example, exosome-like nanovesicles derived from apples are enriched in proteins involved in ABA (a plant hormone) signaling, which regulates various physiological responses in plants, including responses to biotic and environmental stresses. Furthermore, miRNAs contained in these nanovesicles, such as mdm-miR858 and mdm-miR156, have

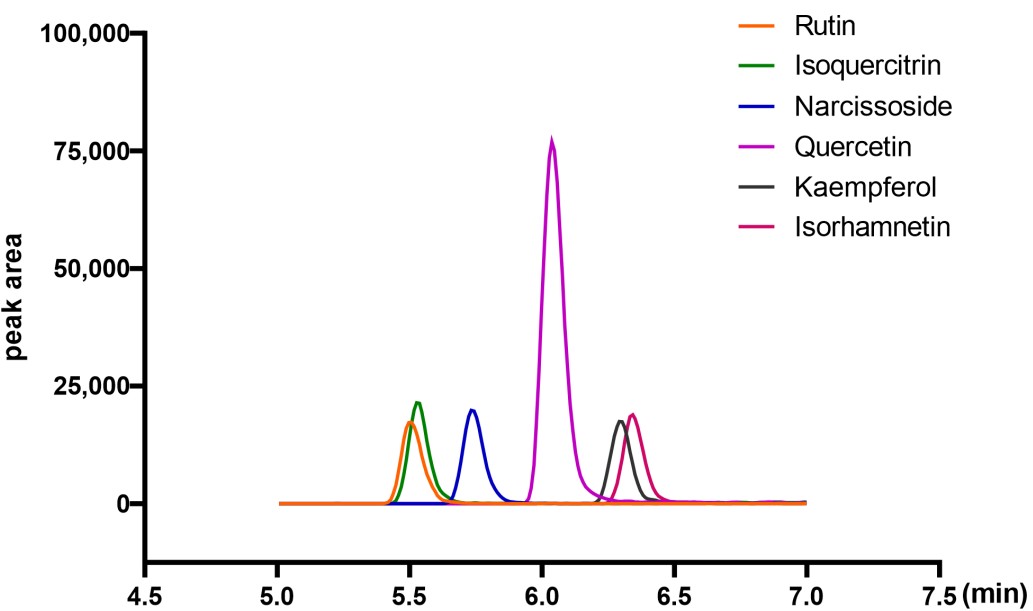

**Figure 3** Representative chromatograms of the extraction of the six analytes (standard substance).

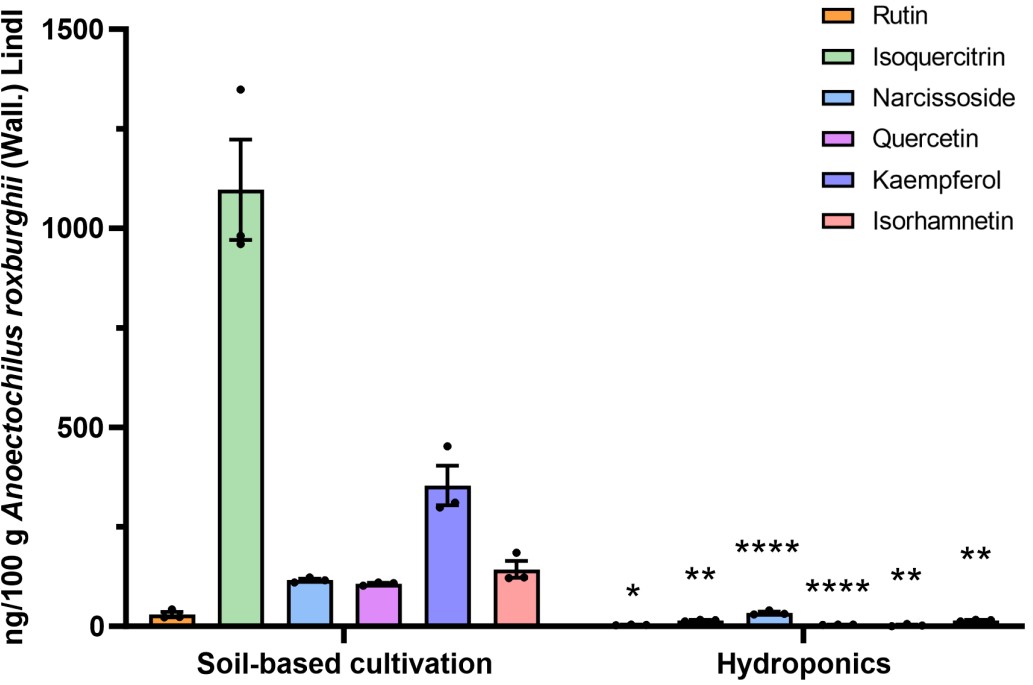

**Figure 4** Respective detection of the six active components in ARENVs isolated from soil-based cultivation and hydroponics. Data are expressed as mean ± SEM. $n = 3$ per group, ng/100 g *A. roxburghii*; * $P < 0.05$, ** $P < 0.01$, **** $P < 0.0001$, compared with the soil-based cultivation group.

regulatory effects on antioxidant activity (*Trentini et al., 2024*). EVs may also participate in plant damage repair. For example, proteomic analyses have shown that *Arabidopsis* callus-derived extracellular vesicles are enriched in proteins associated primarily with stress responses and cell wall modifications (*Yugay et al., 2023*). PENs can also inherit the natural active ingredients of the plant. For example, *G. lucidum*-derived exosome-like nanovesicles are rich in ganoderic acid (*Mi et al., 2025*) and ginseng root-derived exosome-like nanoparticles are abundant in ginsenoside active ingredients (*Choi et al., 2024*). In this study, we used LC-MS/MS to determine that ARENVs are rich in various flavonoid secondary metabolites, including isoquercitrin, kaempferol, isorhamnetin, narcissoside, quercetin and rutin. These important secondary metabolites encapsulated in ARENVs may be transported to other parts of the plant through vesicle transport, where they may play important physiological and defensive roles.

Plant secondary metabolites have a wide range of pharmacological activities. They play an important role in plant response to biotic and abiotic stresses and are also valuable drug resources (*Lv et al., 2024*; *Ramakrishna & Ravishankar, 2011*). Of them, quercetin, a powerful antioxidant, can induce proper plant growth, and can help plants tolerate several biotic and abiotic stresses (*Singh et al., 2021*). In preclinical experiments, quercetin can inhibit oxidative stress, inflammation, and enhance the endogenous antioxidant defense mechanisms (*Alharbi et al., 2025*). It has also been documented to confer neuroprotective and cardiovascular protective effects (*Chiang, Tsai & Wang, 2023*; *Grewal et al., 2021*; *Patel et al., 2018*), and relieve metabolic syndromes such as high blood sugar levels, hyperlipidemia, and obesity (*Hosseini et al., 2021*). As the most common glycoside form of quercetin (quercetin-3-O-glucoside), isoquercitrin exerts multiple chemoprotective effects on oxidative stress, cancer, cardiovascular disease, diabetes and allergic reactions *in vitro* and *in vivo* (*Valentova et al., 2014*). A 3-O-rutinoside derivative of quercetin, rutin has a variety of pharmacological activities, such as anti-inflammatory (*Wang et al., 2023*), anti-aging (*anon, 2025*), anti-ulcer (*Akash et al., 2024*), antihyperglycemic property, and protective effects against the development of diabetic complications (*Ghorbani, 2017*). As an ideal tumor targeting ligand, rutin can target glucose transporters that are highly expressed in various malignant tumors and exhibit photothermal effects after complexing with metal ions (*Fu et al., 2024*). Still, isorhamnetin, a 3-O-methylated derivative of quercetin, can exert anti-tumor and anti-inflammatory effects, and can offer organ protection and obesity prevention by regulating signaling pathways such as PI3K/AKT/PKB, NF-κB, and MAPK (*Gong et al., 2020*). As a 3-O-rutinoside of isorhamnetin, narcissoside mainly confers antioxidant and anti-apoptotic effects (*Hao et al., 2025*; *Fu et al., 2022b*). Finally, kaempferol, a common polyphenol antioxidant, has the capacity to reduce the risk of various chronic diseases such as cancer (*Chen & Chen, 2013*), Alzheimer's disease (*Dong, Zhou & Nao, 2023*), and atherosclerosis (*Chen et al., 2024*).

However, despite the versatile biological activities, the applications of natural quercetin and its derivatives are impeded by their low solubility and bioavailability. Therefore, researchers have developed nanocarriers or structural modifications to improve their utilization (*Alizadeh & Ebrahimzadeh, 2022*; *Alizadeh, Savadkouhi & Ebrahimzadeh, 2023*),

which has brought favorable therapeutic effects. For example, mesenchymal stem cell-derived exosomes loaded with quercetin can treat Parkinson's disease by modulating the inflammatory immune microenvironment (*Zhang et al., 2025*); natural exosomes rich in rutin can be used orally to treat hepatocellular carcinoma without obvious toxicity (*Gao et al., 2024*). Therefore, exosomes can be utilized as a natural carrier to facilitate the bioavailability of such active ingredients. Compared with synthetic nanocarriers (such as polymer nanocarriers (*Nelemans & Gurevich, 2020*), metal and carbon-based particles (*Wolfram et al., 2015*)), PENs have the obvious advantage in terms of safety. In addition, they can provide an effective delivery system not only for their own active ingredients, but also for drugs (*Langellotto et al., 2025*). For example, in the dextran sulfate sodium (DSS)-induced mouse colitis model, methotrexate carried by grapefruit-derived nanovesicles can selectively target lamina propria macrophages and better alleviate intestinal inflammation when compared with the sole treatment of methotrexate (*Wang et al., 2014*). In our study, *A. roxburghii* was found to have exosome-like nanovesicles encapsulating active secondary metabolites, suggesting that ARENVs may have a built-in delivery system for their self-carried active ingredients, which may increase the bioavailability of these secondary metabolites. Furthermore, if exogenous drugs are loaded into ARENVs, their vesicular structure can be leveraged for stable drug delivery, thus producing synergistic pharmacological effects with the intrinsic active ingredients. This combined delivery model of "natural active ingredients + exogenous drugs" may provide a novel approach for the medicinal development of ARENVs exosomes.

In the current study, the contents of six flavonoids in ARENVs were compared between two cultivation systems. The flavonoid content of ARENVs detected in soil-grown *A. roxburghii* was significantly higher than that in hydroponic *A. roxburghii*. This discrepancy may be due to the varying accumulation of secondary metabolites in *A. roxburghii* under different cultivation patterns. *A. roxburghii* roots can directly absorb the single nutrient (chemically pure) in hydroponic nutrient solutions, achieving high absorption efficiency but with a limited composition. Soil, on the other hand, is a natural complex nutrient reservoir, containing not only basic nitrogen, phosphorus, and potassium, but also a rich reservoir of trace elements, small organic molecules, and microorganisms. Roots absorb nutrients through interactions with soil particles and microorganisms, potentially stimulating the synthesis of more secondary metabolites to adapt to the environment. Slight changes in the soil environment can also cause mild abiotic stresses in plants. For example, moderate nutrient deficiency may limit the growth, despite routine photosynthesis, in which carbohydrates accumulate in the nutrient-deficient plants, thus increasing the C/N ratio within the plant, and are allocated to C-based secondary metabolites (all of the above flavonoids are carbon-based secondary metabolites) (*HERMS, 1992*). In addition, the presence of numerous microorganisms in the soil can regulate the production of plant secondary metabolites (*Lv et al., 2024*). This biotic stress is also an important factor in the production of plant secondary metabolites. Previous literature has reported that endophytic fungi from the *Anoectochilus* and *Ludisia* species have beneficial effects on the synthesis of secondary metabolites in *A. roxburghii* (*Ye et al., 2020*). Taken together, the highly controllable and less stressful hydroponic environment may be more conducive to

the primary metabolism of *A. roxburghii*, which produces substances required for growth, while the potential abiotic and biotic stresses in the soil culture environment may act as a stimulus, promoting the secondary metabolic defense pathways of *A. roxburghii* and producing flavonoids and other secondary metabolites. These factors may account for the significant difference in the levels of ARENVs between the soil-cultured source and the hydroponic one.

It is worth noting that there were no obvious morphological differences between soil-grown and hydroponic plants, indicating that visual inspection is not sufficient for quality assessment. Furthermore, even with the same cultivation method, the quality may fluctuate between different batches of *A. roxburghii*, which highlights the need of strict quality control, such as setting minimum content standards for key bioactive ingredients, to ensure consistency in its medicinal value.

However, this study only preliminarily identified the presence of exosome-like nanovesicles in *A. roxburghii*, detected and compared the contents of flavonoid secondary metabolites in soil- and hydroponic-cultured ARENVs. There are also other limitations. We only quantified six flavonoid secondary metabolites in ARENVs, and did not cover other types of active ingredients (terpenes, phenolic acids, polysaccharides, *etc.*) and proteins, lipids, RNA, etc. that may exist in ARENVs. Therefore, the reported secondary metabolites cannot fully reflect the compositional differences of the vesicles. In addition, we have not sufficiently explored the role of ARENVs in the growth and defense of *A. roxburghii*, let alone the specific mechanism underlying the action of ARENVs in medicinal use. In the future, the biological activity of ARENVs can be verified by combining *in vitro* and *in vivo* experiments; more studies are also awaited to explore the regulatory mechanism of key factors in the cultivation environment on vesicle components, and its application potential as a natural drug carrier to provide a more solid theoretical basis for the in-depth development and application of *A. roxburghii*.

## CONCLUSION

This study successfully extracted and identified nanovesicles with typical exosome characteristics from *A. roxburghii*. The LC-MS/MS analysis revealed that ARENVs contained flavonoid secondary metabolites and that the content in ARENVs derived from soil-cultured *A. roxburghii* was significantly higher than that in hydroponic counterparts. This discovery not only enriches the species coverage of medicinal herbal exosome-like nanovesicles, but also deepens the understanding of the storage of medicinal ingredients of *A. roxburghii*. It also lays the foundation for the further development and standardized cultivation of medicinal resources of *A. roxburghii*, and provides new directions for the potential application of ARENVs.

## ACKNOWLEDGEMENTS

Here, we would like to express our heartfelt gratitude to Wenjie Zhuang for his assistance on the visualization of the work. We thank Hongzhi Huang for helpful discussions and language polishing. We are grateful to Professor Liying Huang and Mr. Zhou Zheng for

their assistance with the standard substances. We also would like to express our gratitude to Mr. Shuyuan Wang, Ms. Lin Zhong, Ms. Sishi Chen and Ms. Yan Hu for their assistance on the TEM, ultracentrifugation and LC-MS/MS analysis. We are grateful to Mr. Li-Song Chen for his guidance on the LC-MS/MS analysis.

### Funding

This work was supported by the Natural Science Foundation of Fujian Province (China) (grant number 2024J01700) and High-level Talents Research Start-up Project of Fujian Medical University (grant number XRCZX2022024) to Xue Mi; and Joint Funds for the Innovation of Science and Technology of Fujian Province (China) (grant number 2023Y9126) to Xinglin Ruan. The funders had no role in study design, data collection and analysis, decision to publish, or preparation of the manuscript.

### Grant Disclosures

The following grant information was disclosed by the authors:
The Natural Science Foundation of Fujian Province (China): 2024J01700.
High-level Talents Research Start-up Project of Fujian Medical University: XRCZX2022024.
Joint Funds for the Innovation of Science and Technology of Fujian Province (China): 2023Y9126.

### Competing Interests

The authors declare there are no competing interests.

### Author Contributions

- Xuanzhe Zhou performed the experiments, analyzed the data, prepared figures and/or tables, and approved the final draft.
- Xinglin Ruan conceived and designed the experiments, performed the experiments, analyzed the data, authored or reviewed drafts of the article, and approved the final draft.
- Xue Mi conceived and designed the experiments, performed the experiments, analyzed the data, prepared figures and/or tables, authored or reviewed drafts of the article, and approved the final draft.

### Data Availability

   The raw measurements are available in the Supplemental Files.

### Supplemental Information

Supplemental information for this article can be found online at http://dx.doi.org/10.7717/peerj.20182#supplemental-information.

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
