# Peer review of "Extraction, identification and component analysis of exosome-like nanovesicles in Anoectochilus roxburghii (Wall.) Lindl"

_PeerJ, doi:10.7717/peerj.20182_

## Round 0.1 · original submission · Major Revisions

Please revise the manuscript based on the reviewers' comments.

**Language Note:** The review process has identified that the English language must be improved. PeerJ can provide language editing services - please contact us at [email protected] for pricing (be sure to provide your manuscript number and title). Alternatively, you should make your own arrangements to improve the language quality and provide details in your response letter. – PeerJ Staff

·

Basic reporting

The manuscript with the title: Extraction, identification and component analysis of exosome-like nanovesicles in Anoectochilus roxburghii (Wall.) Lindl. basically is a report from analytical studies of chemical compositions of the plant. The result is actually interesting, especially to my concern is that the quantity of chemicalsinvestigated in the plant gown in soil is much higher that its quantity in plant grown in hydrophonic. Unfortunately there is not enough discussion regarding the result such as why this difference in chemical quantitity is happened.

Experimental design

In the section of materials and methods, the manuscript did not mention references used at all, as if the methods conducted in the experiment never been used by other works, or the method is something universal where there is no refence is needed there.

Validity of the findings

The finding is interesting, whoever it need more discussion, i.e. comparison with other works.

Additional comments

The manuscript need more elaborate discussion in comparing the result with other works and explanation why the result showed as mentioned.

Reviewer 2 ·

Basic reporting

The language is clear but could use refinement and a closer edit to catch minor issues in capitalization, proper use of italics, and describing an acronym prior to using it. The figures presented are clear and easy to understand, I did not check every reference but a superficial review of the references revealed only 1 that I recommended removing. One aspect of their Results (hydroponic vs soil grown plants) was not described in the introduction or methods sections, and needs to be addressed, but it was clear why they analyzed both types in the Results.

Experimental design

No comment, the experimental design was thorough for the scope of this paper.

Validity of the findings

No comment.

Additional comments

This was an interesting manuscript, I enjoyed reading the authors' work and I look forward to seeing what can be done with A. roxburghii in the future.

Annotated reviews are not available for download in order to protect the identity of reviewers who chose to remain anonymous.

---

## Round 0.2 · accepted · Accept

Based on the reviewers' comments, the manuscript can be accepted for publication by the journal.

Reviewer 2 ·

Basic reporting

The readability of the manuscript has been greatly improved, I have no suggestions.

Experimental design

No comment

Validity of the findings

No comment

Additional comments

Some (very) minor edits:
Line 97: space after the 4 in 4*C
Line 32: “Chromatographic and mass spectrometry conditions were set following…”
Lines 277 & 289: I’m not sure if these two instances of “” are necessary. If they are quotes from an external publication then they should be cited, but I think these are neologisms so citations are not necessary…I don’t think the terminology is so unique that it requires the “”…if the authors enjoy their inclusion then please disregard. I agree with the usage for lines 280/281.
Line 297: uncapitalize “All”